# Is There an Opportunity to De-Escalate Treatments in Selected Patients with Metastatic Hormone-Sensitive Prostate Cancer?

**DOI:** 10.3390/cancers16132331

**Published:** 2024-06-26

**Authors:** María Antonia Gómez-Aparicio, Fernando López-Campos, David Buchser, Antonio Lazo, Patricia Willisch, Abrahams Ocanto, Paul Sargos, Mohamed Shelan, Felipe Couñago

**Affiliations:** 1Department of Radiation Oncology, Hospital Universitario de Toledo, 45007 Toledo, Spain; mariang.aparicio@gmail.com; 2Department of Radiation Oncology, Hospital Universitario Ramón y Cajal, 28034 Madrid, Spain; 3Department of Radiation Oncology, Hospital Universitario San Francisco de Asis and Hospital Vithas La Milagrosa, GenesisCare, 28002 Madrid, Spain; abraham.ocanto@gmail.com (A.O.); fcounago@gmail.com (F.C.); 4Department of Radiation Oncology, Hospital Universitario Cruces, 48903 Barakaldo, Spain; davidbuchser@gmail.com; 5Department of Radiation Oncology, Hospital Universitario Virgen de la Victoria, 29010 Malaga, Spain; anlaprados@gmail.com; 6Department of Radiation Oncology, Hospital Meixoeiro, 36214 Vigo, Spain; patriciawillisch@gmail.com; 7Department of Radiation Oncology, Institut Bergonié, 33000 Bordeaux, France; p.sargos@bordeaux.unicancer.fr; 8Department of Radiation Oncology, Inselspital, Bern University Hospital and University of Bern, Switzerland; mohamed.shelan@insel.ch

**Keywords:** metastatic hormone-sensitive prostate cancer, androgen receptor signalling inhibitors, treatment intensification, de-intensification

## Abstract

**Simple Summary:**

First-line treatment options for patients with hormone-sensitive metastatic prostate cancer (mHSPC) have evolved in recent years with treatment intensification strategies used to improve survival and delay disease progression. This study reviews the evolution of treatment intensification in these patients, as well as ongoing trials that will provide us with answers to different questions that we ask in routine clinical practice.

**Abstract:**

The treatment landscape for metastatic hormone-sensitive prostate cancer continues to evolve, with systemic treatment being the mainstay of current treatment. Prognostic and predictive factors such as tumour volume and disease presentation have been studied to assess responses to different treatments. Intensification and de-escalation strategies arouse great interest, so several trials are being developed to further personalize the therapy in these populations. Is there an optimal sequence and a possible option to de-intensify treatment in selected patients with a favourable profile? This and other goals will be the subject of this review.

## 1. Introduction

Prostate cancer continues to be the most prevalent neoplasm in men worldwide [1]. At least 5% of diagnosed cases debut with a metastatic disease, though the vast majority progress from locally advanced stages to disseminated disease following the administration of curative treatment [2]. First-line treatment options for patients with metastatic hormone-sensitive prostate cancer (mHSPC) have evolved in recent years with various treatment intensification strategies used to improve survival, delay disease progression, and enhance quality of life. Leveraging results from trials that combine androgen deprivation therapy (ADT) with androgen receptor signalling inhibitors (ARPIs) or docetaxel with ADT and ARPIs, we have been able to optimize outcomes in the clinical setting, becoming the new standard of care (SOC) [3,4,5,6]. In the same way, radiotherapy to the primary tumour is considered by main international clinical guidelines as a complementary treatment option in selected patients, contributing possible improvements in overall survival for patients with a low tumour burden according to CHAARTED criteria, and controlling the onset of severe urinary toxicity [7]. However, despite these increasingly effective treatment intensification strategies in mHSPC, the permanent remission of the disease and the possibility of discontinuing systemic treatment remain issues to be resolved: a singular approach to treatment intensification in mHSPC patients likely results in overtreatment for a specific subgroup, exposing them to additional therapies and their associated toxicities, which may not be necessary for treating a disease with a more indolent course. As the treatment landscape enriches an earlier stage of the disease, future studies must elucidate biomarkers to define which patients will benefit most from the intensification and/or de-escalation of the systemic treatment, with which agents, and what should be the duration of these treatments.

## 2. Why Intensify Treatment in mHSPC?

The concept of treatment intensification in patients with mHSPC arose in the 1990s, with the publication by the EORTC of two-phase III trials that compared the combination of a luteinizing hormone-releasing hormone (LHRH) agonist with another agent versus standard monotherapy. While the EORTC 30843 [8] trial found no significant differences in survival, response rates, and time to progression, the trial published by Denis et al. showed positive results in favour of the combination when compared to bilateral orchiectomy [9]. 

With the publication of the CHAARTED, STAMPEDE, and LATITUDE [5] trials, interest was rekindled in intensifying treatment in this group of patients. Thus, following publication of the CHAARTED trial and later with STAMPEDE, docetaxel became part of the standard of care in these patients. Sweeney et al. showed that the addition of docetaxel to ADT increased overall survival (OS) compared to ADT alone (57.6 vs. 44 months, *p* < 0.001) as well as progression-free survival (PFS) (20.2 vs. 11.7 months, *p* < 0.001) [10]. In a subsequent analysis, it was observed that the benefit was greater in patients with a high volume of disease, defined as the presence of visceral metastases or ≥4 bone lesions with ≥1 outside of vertebral bodies and the pelvis (HR: 0.63; *p* < 0.001). The OS results in the low-volume subgroup did not reach statistical significance (HR: 1.04; *p* = 0.86) [11]. Then, Clarke et al. confirmed the OS benefit of the combination of docetaxel with ADT over ADT alone, showing no differences based on the volume of metastatic disease in long-term survival results from the STAMPEDE trial [12].

The LATITUDE trial, which defined the term high-risk in metastatic prostate cancer, led to the publication of other trials with favourable results for various androgen signalling pathway inhibitors over ADT. Survival benefits of abiraterone, apalutamide, and enzalutamide subsequently prompted an early intensification of systemic treatment. Unlike the TITAN [3] and ARCHES [4] trials, which exclusively allowed the prior, non-concurrent use of docetaxel, the ENZAMET trial [13] included a percentage of patients who received triple therapy (enzalutamide+docetaxel+ADT). The outcomes obtained for the combination of enzalutamide+docetaxel+ADT showed OS results similar to those of the PEACE-1 and ARASENS trials (HR: 0.73, 95% CI 0.55–0.90) [14]. These last two trials demonstrated a benefit in patients with mHSPC with the intensification of treatment through systemic triple therapy. In the PEACE-1 trial, the combination of abiraterone+docetaxel+ADT+/−radiotherapy to the primary tumour versus docetaxel+ADT+/−radiotherapy to the primary tumour [15] showed a significant improvement in the OS of the triple therapy in patients with a high volume (HR 0.72; *p* = 0.019) but not in patients with a low volume [16]. We must also consider that this trial did not address whether triple therapy is superior to treatment with ADT plus ARPI, a question not covered by the design of the ARASENS trial, which assessed the treatment with darolutamide+docetaxel+ADT versus docetaxel+ADT. The experimental arm showed a higher OS in the overall study population (HR: 0.68; *p* < 0.001); however, given the limited number of patients with metachronous disease (13.9% of the total patients), and as this was not a stratification factor, it prevents us from examining the benefit of triple therapy in this population [6]. 

In this sense, the possibility of intensifying systemic treatment in earlier phases of the disease has also garnered significant interest. The results of the EMBARK trial highlighted that an intensification of treatment with enzalutamide+/−ADT in patients with high-risk biochemical recurrence after prostatectomy+/−post-operative radiotherapy or radiotherapy as a treatment for the primary tumour, reduced the risk of metastasis or death (HR: 0.42; 95% CI, 0.30–0.61; *p* < 0.001) in the combination arm, and (HR: 0.63; 95% CI, 0.46–0.87; *p* = 0.005) in the enzalutamide monotherapy group versus treating these patients without enzalutamide [17]. The results of the PRESTO trial, comparing ADT monotherapy with ADT+apalutamide and ADT+apalutamide+abiraterone in the setting of biochemical recurrence, showed a benefit in PSA progression-free survival in the experimental arms compared to ADT (HR: 0.52 [95% CI: 0.35–0.77]; *p* = 0.00047) for ADT+apalutamide and (HR: 0.48 [95% CI: 0.32–0.71]) for ADT+apalutamide+abiraterone [18]. In the same way, other trials are evaluating the intensification of systemic treatment in patients with high-risk prostate cancer (NCT02531516, NCT06282588, NCT04136353, NCT05826509, and NCT02446444). 

Other strategies under development aim to evaluate agents with different mechanisms of action, some of them currently approved for use in the context of mCRPC, in the clinical setting of mHSPC (Table 1). 

Regardless of the systemic treatment intensification strategy chosen, whether it is dual ARPI+ADT therapy or triple therapy including docetaxel, it remains essential to ensure that patients have equitable access to these therapies in clinical practice, as real-world data suggest that many men with mHSPC continue to be undertreated.

## 3. Current Management of mHSPC

In less than a decade, the therapeutic algorithm for mHSPC has drastically changed with the addition of docetaxel and/or an ARPI to ADT monotherapy. Current efforts are directed at further optimizing mHSPC treatment.

### 3.1. ADT+ARPI

With the publication of results from arm G of the STAMPEDE trial [19], which showed a superior 5-year OS with abiraterone+ADT compared to that with ADT alone (60% vs. 41%) and from the LATITUDE trial [20], in which the combination of abiraterone+ADT outperformed ADT alone in both PFS (HR, 0. 47; 95% CI, 0.39–0.55; *p* < 0.001) and OS (HR 0.66; 95% CI, 0.56–0.78; *p* < 0.0001), with this benefit being statistically significant in patients defined as high-risk [5], results from the use of other ARPIs in this clinical setting began to be published.

Following the publication of the TITAN and ARCHES trials, where apalutamide and enzalutamide demonstrated superiority in OS (HR: 0.65; (95% CI:0.53–0.79) *p* < 0.0001 for apalutamide and (HR:0.66; (95% CI: 0.53–0.81) *p* < 0.001) for enzalutamide) and in rPFS (HR: 0.48, (95% CI 0.39–0.60), *p* < 0.001) and HR = 0.39; (95% CI: 0.30-0.50); *p* < 0.001), respectively, these ARPIs are now part of the current SOC in this patient group.

### 3.2. ADT+Docetaxel+ARPI

PEACE-1 evaluated the addition of abiraterone to docetaxel+ADT, established as the SOC following the publication of the CHAARTED and STAMPEDE trial data, achieving superior outcomes compared to those of the control arm (docetaxel+ADT) in both survival (HR: 0.82, 95.1% CI 0.69–0.98; *p* = 0.030) and rPFS (HR: 0.54, 99.9% CI 0.41–0.71; *p* < 0.0001). Unlike PEACE-1, the ARASENS trial included both de novo and metachronous mHSPC patients, demonstrating the superiority of the triple regimen (darolutamide+docetaxel+ADT) over docetaxel+ADT in patients with a high disease volume as per CHAARTED criteria, high- and low-risk per LATITUDE criteria [21], although it did not address in its design the potential role of radiotherapy on the primary tumour [6].

To date, no studies have directly compared systemic triple therapy against apalutamide/enzalutamide+ADT; however, we do have results from meta-analyses and indirect systematic reviews that shed some light, albeit with numerous limitations, on which treatment strategy is superior [22,23,24,25,26]. In this respect, triple therapy is superior to docetaxel plus ADT in terms of OS and rPFS. However, it is not superior when compared to ARPI+ADT.

### 3.3. Radiotherapy to the Primary Tumour

The PEACE-1 trial included an evaluation of radiotherapy on the primary tumour. Following the data published from the H arm of STAMPEDE [7], which showed that radiotherapy on the primary tumour increased OS (HR: 0.68; *p* < 0.007) and PFS (HR: 0.59; *p* < 0.0001) in patients with a low volume receiving ADT alone, and in the STOPCAP meta-analysis [27], primary treatment was a part of trial. Recent results have shown that the addition of prostate radiotherapy did not correlate with improvements in OS, neither in the SOC group (HR: 1.18, 95% CI: 0.81–1.71, *p* = 0.39) nor in the SOC+abiraterone group (HR: 0.77, 95% CI: 0.51–1.16, *p* = 0.21), but it did result in a better rPFS outcome in the experimental arm compared to that in the SOC group (median 7.5 versus 4.4 years, *p* = 0.02) [28]. Prostate treatment was also associated with increased castration-resistant survival in the low-volume group (HR: 0.74, 95% CI: 0.60–0.92, *p* = 0.007) and improved time until severe genitourinary events across the entire study cohort (*p* < 0.001). Therefore, RT on the prostate only in patients with low-volume metastatic disease should be considered.

### 3.4. Metastasis-Directed Therapy

In addition to the above, the role that metastasis-directed therapy can play in the management of patients with synchronous or metachronous oligometastatic disease is substantial. There are data from phase II trials that demonstrate the efficacy and safety of SBRT, especially in metachronous patients, although without concurrent treatment with ARPIs and with a design that did not establish this approach as the standard of treatment [29,30,31,32]. In this regard, phase III trials assessing the role of MDT in oligometastatic or oligorecurrent patients are ongoing (Table 2).

Therefore, with all this information, the treatment table for patients with mHSPC could be as follows (Table 3):

## 4. De-Intensification of Treatment: Is There a Reason to Consider It?

Before the emergence of systemic treatment intensification in these patients, intermittent ADT was based on the following benefits: on one hand, delaying progression to castration resistance and, on the other, the probable improvement in quality of life by minimizing the adverse health effects of continuous castration. The SWOG 9346 trial evaluated the role of intermittent ADT therapy compared to continuous treatment in patients with mHSPC who achieved a PSA < 4 ng/mL in the first 7 months. The results showed no inferiority in OS for intermittent ADT, although there was an improvement in quality of life in that arm, leading to heterogeneous clinical practice in this regard [33]. 

Nowadays, there is growing interest in identifying response-based criteria that can guide not only at a prognostic level but also at a predictive level the development of de-intensification strategies for patients with favourable profiles. In this context, PSA is the most studied assessment criterion. In various trials, the decrease of PSA in a short period has been associated with an increase in OS and PFS. Thus, in the LATITUDE trial, both the depth and durability of PSA responses measured by the PSA50 response, the PSA90 response, and the reduction to PSA concentrations ≤ 0.1 ng/mL in ≤6 months correlated favourably with the long-term study outcomes in terms of rPFS and SG. Likewise, in an exploratory analysis of TITAN, the association of a profound decrease in PSA in prognostic terms was evaluated [34]. At 3 months of treatment with apalutamide, a profound PSA decrease of ≥90% or to ≤0.2 ng/mL occurred in 59% and 51% of cases in the apalutamide group and in 13% and 18% in the placebo group, respectively. Achieving a profound PSA decrease at 3 months was associated with a longer OS, higher rPFS, longer time to PSA progression, and more extended time to castration resistance in the apalutamide group compared to those of the placebo (*p* < 0.0001). Recently published real-life data have shown that achieving a PSA nadir ≤ 0.2 ng/mL at any point during treatment is associated with a statistically significant improvement in rPFS and OS (*p* < 0.001) [35]. 

All this leads to considering the possibility of de-escalating treatment in selected patients due to the implications of continuous treatment: treatment side effects, impacts on quality of life, and costs to health systems. 

Currently, the A-DREAM trial, a phase II study, is evaluating the cessation of systemic treatment at 18–24 months from the start of treatment, both of ADT and ARPI, in patients who achieve a PSA < 0.2 ng/ml. The goal is to maximize patients’ quality of life during the course of the disease (NCT05241860). The EORTC-2238 GUCG (de-escalate) trial [36] is a phase III trial that is assessing the role of intermittent ARPI+ADT in patients who achieve a PSA decline to <0.2 ng/mL in the first 6–12 months of treatment compared to continuous treatment. The aim is to show the non-inferiority of OS in patients with discontinuous treatment and to demonstrate that the proportion of patients who do not need to restart their treatment after the first year of suspension is not less than 70%. Finally, another trial is evaluating the intermittency of ADT with apalutamide (NCT05884398). In the first 6 months, all patients receive apalutamide+ADT. After that time, patients who achieve a PSA < 0.2 ng/mL are randomized into intermittent versus continuous ADT, both arms receiving apalutamide. The co-primary objectives are rPFS and the severity of hot flashes at 18 months post-randomization. In this context, the EMBARK trial already evaluated the monotherapy of enzalutamide without ADT in patients with high-risk biochemical relapse, leaving its role yet to be defined [17].

Given the increasing interest in studying the de-escalation of systemic treatment in these patients, the PEACE-6 trial plans to open two arms to assess the de-escalation of systemic treatment in patients with a good PSA response at 6–8 months from the start of treatment or, conversely, to intensify treatment in those who do not have it. The NCT06177015 is evaluating the intermittency of darolutamide in patients receiving triplet therapy. At 6 months of treatment, those with a PSA < 0.2 ng/dL or a PSA > 0.2 but with a decrease of more than 90% relative to the initial value will be randomized to continuous treatment with darolutamide+ADT or ADT alone in the intermittency arm. The primary objectives are rPFS and OS. 

Based on the future results of these trials, it will be necessary to evaluate which approach is most appropriate: the intermittency of ADT and/or ARPI versus SOC in terms of efficacy, quality of life, and economic cost (Table 4).

## 5. The Role of Clinical and Molecular Biomarkers in Decision Making

In both intensification treatment and de-escalation, we need clinical biomarkers to help stratify risk, providing a framework to guide initial treatment selection and when to intensify or de-escalate treatment. In the precision medicine era, they will allow for greater treatment personalisation, as well as the possibility of administering targeted therapies. These biomarkers can be predictive or prognostic, clinical or molecular. 

### 5.1. Clinical Biomarkers

Numerous studies have attempted to identify clinical variables, such as characteristics of the disease or the patient, which act as reliable prognostic factors, indicating either aggressive or indolent tumour biology, and helping to predict likely disease progression.

The presentation of the disease, whether de novo or metachronous, marks the patients’ progression. Various data have indicated a poorer prognosis for those diagnosed with metastatic disease at onset compared to those presenting with recurrent disease after primary tumour control [37,38,39]. The stratification of the disease by volume/risk has also provided information about disease progression. Patients classified as high-risk according to LATITUDE criteria [5] or high-volume according to CHAARTED [10] criteria are associated with lower OS [40]. Not only the volume of the disease influences outcomes, the location and number of metastases must also be considered. Bone lesions are the most common and are associated with worse survival when located outside the pelvis and spine [41]. M1a disease has a more indolent course than visceral metastases [42,43]. Within this patient subgroup, the latest data indicate that secondary lesions located in lungs show survival similar to bone disease and better outcomes than liver or brain metastases [44].

PSA levels and Gleason score can provide information about the type of disease we are facing. We know that serum PSA increases with disease progression, and a rapid kinetics potentially indicates greater aggressiveness [45,46,47]. On the other hand, the speed of response (reaching the nadir before 3–6 months) and its depth (a PSA less than or equal to 0.2 ng/mL or a decrease of ≥90%) are prognostic of higher OS [34,48,49]. A predominant pattern 5 in Gleason grade seems to negatively influence disease progression [5,50,51]. However, the results of various trials show no differences in survival based on it [3,10,14,21,52].

Markers related to the patient, age, performance status (PS), and nutritional status must be considered when selecting a treatment and, therefore, can modify the course of the disease. Regarding age, data from major trials found no differences in OS when the population was analysed by subgroups but observed poorer tolerance and greater treatment discontinuation in patients over 70 years old [13,14,53,54]. These results agree with the real-world data [55,56]. However, patients under 60 appear to present a more aggressive disease with early relapses [57]. There are contradictory data regarding PS. While some studies have shown that a PS above 1 is associated with a worse prognosis, others have described that systemic treatment in these patients reduces the risk of death, although the results are more favourable in patients with a better PS [58]. Sarcopenia appears to be an independent prognostic factor for predicting rPFS and time to PSA progression; therefore, several recent observational studies have indicated that physical activity is beneficial in preventing disease recurrence and improving OS [59].

Other biomarkers, such as perineural invasion [60], anaemia [61], elevated serum lactate dehydrogenase levels [62,63], elevation in alkaline phosphatase (related to bone metastases) [64], low testosterone levels prior to treatment [65], or the presence of symptoms derived from metastases [66], have been shown to be negative prognostic factors.

There is great variability in the prognosis of patients with prostate cancer, despite the advances that have been made in the diagnosis and treatment of these patients. In recent years, several studies that have delineated the genetic landscape of prostate cancer using sequencing techniques have demonstrated the presence of alterations in biologically and clinically relevant pathways [67,68], allowing for better classification of tumours into prognostic groups and laying the groundwork for the development of personalised treatments. Future studies with more ambitious inclusion criteria, as well as more real-world data, will be necessary to determine the clinical impact of these biomarkers more accurately.

### 5.2. Molecular Biomarkers

The identification, development, and validation of molecular markers as decision-making tools in prostate cancer have primarily focused on the context of localised disease [69,70,71] and castration-resistant advanced disease. However, the existence of such analyses in the scenario of metastatic hormone-sensitive prostate cancer are scarce (NCT03413995, NCT04493853, NCT04497844, and NCT04126070).

Nonetheless, we are increasingly understanding the molecular profile of mHSPC. Recently, Van der Eecken et al. [72] published a systematic review that included data from 11 cohorts and over 1600 patients describing the mutational landscape of mHSPC in somatic and/or germline samples. According to this study, the most frequently mutated genes are TP53 (32%) and PTEN (20%). Alterations in cell cycle signalling vary between 7% and 13%, while deletions of RB1 were observed in 6% of cases. Alterations in the DDR (DNA Damage Response) pathway appear in 18% of cases, with BRCA2 being the most frequently altered gene within this group (7%). Lastly, alterations in the androgen receptor pathway were observed only in tumours from patients already treated with androgen suppression therapy. Moreover, it appears that the molecular profile is related to well-established clinical prognostic variables in mHSPC: disease burden (high vs. low) and the timing of metastasis appearance (synchronous vs. metachronous). Thus, alterations in TP53 (35% vs. 29%), BRCA2 (10% vs. 4%), PIK3CA (8% vs. 2%), RB1 (7% vs. 3%), and APC (11% vs. 9%) are more frequent in tumours with a high volume compared to those with a low volume. Regarding the timing of metastasis appearance, more alterations are observed in cell cycle signalling pathways, the Wnt pathway, PTEN, and SPOP in patients with metachronous disease compared to those with de novo metastatic disease. Conversely, alterations in CDK12 (6% vs. 1%) and FOXA1 (17% vs. 10%) are more prevalent in de novo metastatic disease, while alterations in ATM and RB1 are equally prevalent in both groups. 

In the specific context of oligometastatic mHSPC, two studies have demonstrated the prognostic value of a high-risk genomic signature (HiRi) that includes the assessment of somatic mutations in ATM, BRCA1/2, Rb1, and TP53 through next-generation sequencing (NGS). On one hand, Deek et al. [32] conducted a joint analysis of patients included in the ORIOLE and STOMP trials, two randomized phase II trials that compared metastasis-directed therapy (MDT) against observation in patients with oligometastatic mHSPC and demonstrated an improvement in PFS in the experimental arms. In this analysis of 70 patients, the presence of high-risk mutations was significantly associated with worse rPFS regardless of the treatment received—10 vs. 22.6 months (HR 0.38; 95% CI, 0.20–0.17; *p* = 0.01). Moreover, in an analysis stratified by treatment (MDT vs. observation), it was confirmed that both patients with HiRi mutations and those without them benefited from MDT, but the magnitude of the treatment effect appeared greater in patients with high-risk mutations (HR 0.05; 95% CI 0.01–0.28; *p* = 0.01 for patients with HiRi mutation vs. HR: 0.42; 95% CI 0.23–0.77; *p* = 0.01). On the other hand, Sutera et al. [73] evaluated the prognostic value of this same signature in a cohort of 101 patients with de novo mHSPC, a low disease burden, and its association with disease control outcomes in those patients who received primary tumour radiotherapy. In this case, patients presenting with a HiRi mutation did not appear to benefit from primary irradiation, whereas those without mutations showed significantly better rPFS, time to castration resistance, and OS.

Although the results of these studies may seem contradictory, it is important to consider that they are exploratory analyses of different patient cohorts, at different stages of the disease (synchronous vs. metachronous), with different treatments (MDT vs. RT on the primary tumour), and different comparator arms (observation vs. active treatment). Nonetheless, it appears that this high-risk genetic signature is a potential predictor of response to different treatment strategies in mHSPC and should be validated in prospective cohorts.

## 6. Conclusions

Advancements in the treatment of mHSPC in recent years have allowed us to have multiple treatment strategies in this clinical scenario. However, we need molecular and clinical biomarkers that enable us to break the myth of personalized treatment in this phase of the disease, optimizing the results obtained to date.

## Figures and Tables

**Table 1 cancers-16-02331-t001:** Current trials evaluating novel therapies and treatment combinations in patients with mHSPC.

Trial	Phase	Experimental Arm	Disease Group	Primary Outcome
NCT04343885	II	LuPSMA+docetaxel	De novo, high-volume mHSPC	Undetectable PSA at 12 months
NCT04443062	II	LuPSMA	Oligometastatic mHSPC	Disease progression after 6 months
NCT04748042	II	Olaparib+abiraterone+ADT+SABR	Oligometastatic mHSPC	Percentage of patients without failure after 24 months
NCT04262154	II	Atezolizumab+abiraterone+ADT+SABR	De novo mHSPC	Two-year failure rate
NCT03795207	II	Durvalumab+SABR	Relapsed low-volume mHSPC (visible on PET scan only)	Two-year progression-free survival
NCT06312670	II	Combining EPI-7386+enzalutamide+ADT	De novo, low volume	Biochemical response rate
NCT03951831	II	Combined hormonal chemoimmunotherapy (REGN2810)+docetaxel	De novo mHSPC	Undetectable PSA at 6 months
NCT04126070	II/III	Nivolumab+ADT+docetaxel in DNA damage repair defects	mHSPC	PSA decline to <0.2 ng/mL at 7 months
NCT03879122	II/III	Immunotherapy+docetaxel+ADT	De novo, high volume	OS
NCT06392841	II/III	Niraparib, abiraterone acetate and prednisone with HRR alterations	De novo mHSPC	PSA decline to <0.2 ng/mL at 24 weeks
NCT05956639	III	6-month vs. Long-term Course of Rezvilutamide with ADT+Chemotherapy	De novo, high volume	Radiological progression free survival (rPFS) at 36 months
NCT04821622	III	Talazoparib With enzalutamide in men with DDR gene-mutated mCSPC	De novo mHSPC	rPFS
NCT04720157	III	177Lu-PSMA-617+ARPI+ADT	De novo mHSPC	rPFS

**Table 2 cancers-16-02331-t002:** Current phase III trials evaluating stereotactic body radiation therapy (SBRT) as metastasis-directed therapy in the context of treatment intensification.

Trial	Phase	Experimental Arm	Disease Group	Primary Outcome
NCT05209243	III	SBRT plus standard of care in castration sensitive oligometastatic prostate	Oligometastatic prostate carcinoma	rPFS
NCT04115007	III	Standard of care + SBRT	Oligometastatic prostate cancer	Castration-resistant prostate-cancer-free survival
NCT05352178	III	Addition of short-term androgen deprivation therapy (ADT) for 1 month or short-term ADT for 6 months together with an androgen-receptor-targeted therapy (ARTA) to metastasis-directed therapy (MDT)	Oligorecurrent hormone sensitive prostate cancer.	Poly-metastatic-free survival
NCT04787744	III	Standard systemic therapy with or without PET-directed local	Oligometastatic prostate cancer	Castration-resistant prostate cancer-free survival
NCT04983095	III	MD–SBRT in addition to standard treatment	Oligometastatic prostate cancer	Failure-free survival

**Table 3 cancers-16-02331-t003:** mHSPC treatment algorithm based on tumour burden and disease presentation.

Disease Type	ADT	AR Pathway Inhibitor	Docetaxel+ARPI	Prostate RT	MDT
**De novo, high volume**	Suboptimal treatment	YES	YES	+/− (Symptom control)	NO
**De novo, low volume**	Suboptimal treatment	YES	+/− Individualize	YES	+/− (no OS data)
**Metachronic disease, high volume**	Suboptimal treatment	YES	YES	NO	NO
**Metachronic disease, low volume**	Suboptimal treatment	YES	+/− Individualize	NO	+/− (no OS data)

**Table 4 cancers-16-02331-t004:** Possible treatment strategies in terms of efficacy, quality of life, and economic cost.

Treatment Strategies	Quality of Life Benefit	Fewer Adverse Effects	Fewer Economic Cost
Maintain ADT+ARPI continuous	**SOC**		
De-escalate by removing ADT	¿?	+	+
De-escalate by removing ARPI	¿?	++	+++
De-escalate by removing all	¿?	+++	++++

¿?: Quality of life benefit is unknow. + refers to the degree of adverse effects or cost that the therapeutic maneuver would have.

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
