# Peer review of "Is There an Opportunity to De-Escalate Treatments in Selected Patients with Metastatic Hormone-Sensitive Prostate Cancer?"

_cancers, 2024, doi:10.3390/cancers16132331_

Round 1

Reviewer 1 Report

Comments and Suggestions for Authors

Comments regarding the Review titled: Is there an opportunity to de-escalate treatments in selected patients with metastatic hormone-sensitive prostate cancer?

In this review, the authors present the changes that have been introduced to the clinical management of metastatic hormone-sensitive prostate cancer (mHSPC) during the past years. They also present the various ongoing clinical trials focused on mHSPC along with molecular and clinical markers that could be useful for disease monitoring and treatment selection. The authors make a case in favor of de-escalating treatments in this patient population.

Some comments:

On table 3, I would change the structure of the table by including the disease type in the first column and the treatment options in the first row. Then, the table will be more focused in the disease types instead of the treatment options.

Table 4 does not have a title. Please check for typos. Maybe label the first column as Treatment strategies.

Line228: Cites table 2, it should be table 4.

In the Conclusion, the authors state the “need for molecular and clinical biomarkers that enable us to break the myth of personalized treatment in this phase of the disease, optimizing the results obtained to date”. However, in line 277, the authors recognize the importance of clinical markers due to the heterogeneity of mHSCP and their usefulness for designing personalized treatments. Please elaborate on the topic of personalized treatments vs disease heterogeneity.

Reviewer 2 Report

Comments and Suggestions for Authors

The manuscript entitled “Is there an opportunity to de-escalate treatments in selected patients with metastatic hormone-sensitive prostate cancer?” is an excellent review paper concerning the treatments currently used in clinical trials designed to improve the condition of patients affected by metastatic hormone-sensitive prostate cancer. The authors critically and purposefully analyze the various clinical trials, mainly concerning those involving the co-administration of 2 or more drugs, and propose various alternatives for reducing treatment doses, stratification of patients by new biomarkers etc. Overall, This study is particularly meticulous and the results are interesting. It is well written and very didactic and I do not find any significant incorrectness. However, some additions and modifications will significantly improve the quality of the article:

- Being aware that the manuscript's topic is clinical trials, the author should also mention and discuss published non-clinical but very promising studies. One possibility might be to add a new, even short, chapter concerning basic and/or preclinical research studies. Alternatively, some of these studies could be integrated into chapters 4 or 5. Some examples are the study DOI: 10.1038/s41419-020-03256-5, regarding drug targeting of locally advanced prostate cancer, and DOI: 10.1007/s11033-024-09506-5; DOI: 10.1038/s41598-021-02675-4; DOI:10.1093/carcin/bgz129; DOI: 10.3389/fonc.2016.00024….and several others.

- The authors should also improve the quality of tables and check the whole text for typographical errors.

Round 2

Reviewer 2 Report

Comments and Suggestions for Authors

Dear Dr López Campos,

you and the other authors have addressed all my suggestions. I found your responses quite satisfactory and the revised version has improved greatly. I now recommend the paper for publication in Cancer

Best Regards